# Oxone^®^-Mediated TEMPO-Oxidized Cellulose Nanomaterial Ultrafiltration and Dialysis Mixed-Matrix Hollow Fiber Membranes

**DOI:** 10.3390/polym12061348

**Published:** 2020-06-15

**Authors:** John P. Moore, Kristyn Robling, Cristian Romero, Keturah Kiper, Soma Shekar Dachavaram, Peter A. Crooks, Jamie A. Hestekin

**Affiliations:** 1Ralph E. Martin Department of Chemical Engineering, University of Arkansas, Fayetteville, AR 72701, USA; jpm011@uark.edu (J.P.M.II); knrobling@gmail.com (K.R.); ciromero26@gmail.com (C.R.); kkiper@purdue.edu (K.K.); 2Department of Pharmaceutical Sciences, University of Arkansas for Medical Sciences College of Pharmacy, Little Rock, AR 72205, USA; SSDachavaram@uams.edu (S.S.D.); PACrooks@uams.edu (P.A.C.)

**Keywords:** ultrafiltration, dialysis, cellulose, nanomaterial, TEMPO, clearance, flux

## Abstract

Recent exploration of cellulose nanomaterials has resulted in the creation of Oxone^®^-Mediated TEMPO-Oxidized Cellulose Nanomaterials (OTO-CNMs). These materials, when incorporated into a polymer matrix, have properties showing increased flux, decreased membrane resistance, and improved clearance, making them an ideal material for dialysis. This study is the first to focus on the implementation of OTO-CNMs into hollow fiber membranes and a comparison of these membranes for ultrafiltration and dialysis. Ultrafiltration and dialysis were performed using bovine serum albumin (BSA), lysozyme, and urea to analyze various properties of each hollow fiber membrane type. The results presented in this study provide the first quantitative evaluation of the clearance and sieving characteristics of Oxone^®^-Mediated TEMPO-Oxidized Cellulose-Nanomaterial-doped cellulose triacetate mixed-matrix hemodialyzers. While the cellulose nanomaterials increased flux (10–30%) in ultrafiltration mode, this was offset by increased removal of albumin. However, in dialysis mode, these materials drastically increased the mass transfer of components (50–100%), which could lead to significantly lower dialysis times for patients. This change in the performance between the two different modes is most likely due to the increased porosity of the cellulose nanomaterials.

## 1. Introduction

According to the Centers for Disease Control and Prevention (CDC), chronic kidney disease is the 14th leading cause of death in the United States [1]. There are an estimated eight million American adults with reported diagnoses of kidney disease [2]. The CDC found that, in 2016, over 50,000 Americans died from kidney disease [3,4]. Recent research has shown renal replacement therapy is essential in treating COVID-19 [5]. Currently, the most common treatment technique is to filter the blood through hemodialysis or peritoneal dialysis. Dialysis does not cure the disease, like kidney transplantation. Still, it can remove contaminants from the blood, like the kidneys, to maintain the health of the individual for a significant period. The majority of patients will receive treatment three times per week, for 3 to 6 h. Each treatment is based on how much blood can be filtered out of the body at a specific time, as well as the performance of the membranes.

Hemodialysis machines operate on the principle of diffusion [6]. Blood is pumped out of the body and passed through hollow fiber modules. The membranes are made of a semi-permeable material, usually cellulose or polysulfone, that has a pore size that is too small to allow blood cells and large proteins to pass through the membrane [7,8]. A solution containing salts and proteins, called the dialysate, is passed on the other (shell) side of the membrane. A lack of urea in the dialysate causes the toxin to be pulled through the filter quickly. Additionally, if there is an excess or absence of proteins or certain salts, such as sodium or potassium, the dialysate can adjust the balance to replace or remove them. After the blood flows by the membrane, it returns to the body. During treatment, patients can also undergo an ultrafiltration process to remove water from the body. Patients often experience fluid retention, so dialysis functions to remove not only excess waste but also excess water. A functional dialysis membrane will allow sizeable-molecular-weight proteins to be retained, smaller proteins and some middle-molecular-weight molecules to pass through in significant amounts, and small compounds like urea, salts, etc., to pass through at almost 100% [9,10]. The most common way to characterize what is passed by the membrane is with a sieving coefficient. A sieving coefficient is a measure of how much will diffuse through the membrane as a function of feed (or blood) concentration. Commercial membranes will typically have a sieving coefficient of albumin ranging from 7 × 10^−4^ to 3 × 10^−3^ [11]. These membranes would usually lead to a loss of less than 3 g of albumin per dialysis session. Sieving coefficients for middle molecules range from 0.1 to 1.0, depending on the type of the membrane [11,12]. Although there are many membranes that can perform dialysis, there is still a need for membranes with higher diffusion potential, reduced treatment times, lower fouling, and little to no potential for clotting [13].

Cellulose is one of the most common types of materials used to create dialysis membranes [14]. Cellulose is very useful for ultrafiltration in dialysis applications and is relatively inexpensive and easy to work with; however, improvements can be made to the material to make dialysis more efficient and improve the outcome of the treatment. Modified cellulose membranes have been shown to decrease the mortality rate of patients undergoing hemodialysis [15]. Although the exact reason for this is not known, it is hypothesized that the modifications allow for more uremic toxins to pass through the membrane while retaining the more significant components in blood. As such, we have proposed the incorporation of cellulose nanomaterials (CNMs) into cellulose membranes for dialysis. Duranti et al. found that cellulosic membranes could be highly desirable for middle molecule removal [16]. Other previous studies using various types of nanoparticles have also shown improvements in anti-protein adsorption and anti-fouling properties [17]. Additional studies have shown that adding TEMPO-oxidized nanocellulose gives the cast material higher tensile strength and flux [18,19]. Furthermore, a non-toxic reaction at mild conditions using TEMPO-oxidation with Oxone^®^ as an intermediary for nanocellulose production yields two forms of Oxone^®^-mediated TEMPO-oxidized cellulose nanomaterials (OTO-CNMs), Form I and Form II [20]. Form I is partially oxidized, and Form II is fully oxidized. They were both found to have a novel crystalline structure [21]. Neither Form I or Form II has been tested within polymer-based filtration studies. Figure 1 shows the general chemical structure of Form I compared to Form II. The production of Form I and Form II was also found to be an efficient method of crystalline nanocellulose production.

The aforementioned results of high flux, increased strength, and anti-fouling capabilities suggest that the untested Form I and Form II nanocellulose particles could cause significantly better performance in a hemofiltration setting. A less-fouling membrane opens up the potential for an implanted dialysis filter, similar to an artificial kidney [22,23]. Additionally, faster clearance creates the opportunity to reduce treatment time [24]. A study in the New England Journal of Medicine found that patients who underwent dialysis with a higher flux rate were shown to experience lower mortality rates by eight percent [25].

This study is the first to focus on the implementation of two derivatives of cellulose nanomaterials into hollow fiber membranes and a comparison of these membranes for ultrafiltration and dialysis. In this work, cellulose nanomaterial mixed-matrix membranes have been evaluated for their potential in dialysis by determining model molecule selectivity (BSA, urea, and lysozyme), as well as transport rates (flux and diffusion rates).

## 2. Materials and Methods

### 2.1. Materials

Utilizing the procedure from Moore et al., OTO-CNMs Form I and Form II were synthesized and used in conjunction with cellulose triacetate (CTA) from Acros Organics (Fair Lawn, NJ, USA), N-methyl pyrrolidone (NMP) from VWR (Radnor, PA, USA), and deionized water to create novel membranes for filtration [21]. The proteins used for ultrafiltration and dialysis characterization were bovine serum albumin (BSA), lysozyme, and urea supplied from VWR. These are common proteins found in the blood. BSA was chosen to represent large proteins in the blood; lysozyme was selected to represent middle molecular weight proteins in the blood, and urea was chosen because it is the primary waste product that is excreted from the kidneys. All membranes were housed in polyvinyl chloride pipes (PVC) supplied from Lowe’s (Fayetteville, AR, USA) using underwater epoxy resin. Ethanol for cleaning was also provided through VWR.

### 2.2. Preparation of Cellulose Solutions

Four different solutions were created for membrane casting. The first, the control, was created using only CTA and NMP. The casting solution was made using 10 wt % CTA and 90 wt % NMP. These were mixed in a bottle and placed on a bottle roller for 5–7 days to allow for the complete incorporation of the CTA. The second solution, Form I, was created using CTA, NMP, and OTO-CNM Form I. The casting solution was made using 9 wt % CTA, 90 wt % NMP, and 1 wt % Form I. To incorporate the Form I nanocrystals, NMP and Form I were blended in a 500-mL beaker at 6000 RPMs for 5 min in a water bath. Then, the solution was placed in a Qsonica (Newtown, CT, USA) sonicator at 500 watts at 20 kHz for 5 min with 20-s intervals of sonication followed by 10 s of no sonication. After sonication, the OTO-CNMs were fully dispersed in solution. The solution was then run through a coarse filter using vacuum filtration to remove any large chunks. After filtration, it was added to a roller bottle with the CTA and placed on a bottle roller for 5–7 days. This process was repeated with the other two types of solutions but with adjusted concentrations of OTO-CNM Form I and Form II. The third solution, 50/50, was prepared using CTA, NMP, and Form I and Form II. The solution was made using 9 wt % CTA, 90 wt % NMP, 0.5 wt % Form I, and 0.5 wt % Form II. The same mixing procedure was followed that was used for Form I. The fourth solution, Form II, was prepared using CTA, NMP, and Form II. The solution was made with 9 wt % CTA, 90 wt % NMP, and 1 wt % Form II. This solution was mixed with the same procedure as Form I and 50/50. All compositions of OTO-CNM/CTA membranes based on wt % for Form I to Form II can be found illustrated in Appendix A.

### 2.3. Membrane Casting

Membranes were created with non-solvent-phase-induced separation hollow fiber casting. A water bath at 35 °C was filled so the fibers would be fully immersed as they traveled the length of the bath. A bore solution containing 15 wt % NMP and 85 wt % deionized water was utilized. A spinneret was used to extrude the hollow fibers. It was set to obtain a 200-µm thickness of the membranes. The polymer solutions were passed through the outer layer of the spinneret at a pressure of 25 psi, and the bore was passed through the spindle at a pressure of 1–2 psi to create a hollow tube. The spinneret was set at a 5-cm height above the water level to obtain a consistent time spent passing into the bath. As the solutions passed through the water bath, phase inversion occurred. The fibers were collected in a roll at a rate of 1.7 m per minute until casting was complete. Immediately after completion, the fibers were placed in a hot water bath at 87–89 °C for 3 min. This heat treatment sets the membranes to strengthen them and ensure high quality. After heat treatment, they were placed in a container filled with room temperature deionized water for storage until further use. No membrane was stored more than one month before use.

### 2.4. Module Assembly

For each dialysis run, a new module had to be created. For each module, PVC pipes were assembled in the fashion shown in Figure 2 to create complete hemodialysis modules for testing. Each module contained 20 fibers. The fibers were inserted into the PVC apparatus and glued into place with underwater epoxy resin. This glue was used to seal the ends of the device, so no leaking would occur during testing. About 3 mL of glue was used for each end. After the glue fully solidified, the ends were cut to create an even surface, and threaded caps were added to the module to create the completed module. These were stored in a room temperature deionized water bath until ready for testing (up to a month).

### 2.5. Dialysis and Ultrafiltration Testing

Dialysis and ultrafiltration module testing began with apparatus assembly. The filtration apparatus allowed solutions to pass through the membrane during ultrafiltration and two fluids to pass concurrently during dialysis. The transmembrane pressures were kept consistent with the use of pressure dampers. Once the module was connected, water was passed through the dialysate side (outside) and the feed side (inside) of the membranes. The system was run for one hour prior to testing at 1–2 psi to stabilize the membrane. The dialysate side was run at a flow rate of 300 mL/min, and the feed side was run at a flow rate of 200 mL/min. The transmembrane pressure was kept consistent at 1 psi. During the hour of stabilization, the feed solutions were prepared. For BSA and lysozyme experiments, the concentration was 1 mg/mL; specifically, 800 mg of protein was dissolved into 800 mL of water. For the urea experiments, the concentration was 2 mg/mL; specifically, 1600 mg of urea was dissolved into 800 mL of water. Once the pure water stabilization was finished, the setup was switched over to ultrafiltration, which involved turning off the dialysate. The feed was maintained at a flow rate of 200 mL/min. Three water samples were collected from the permeate (dialysate side) at 1 psi transmembrane pressure. Each sample was collected for 90 s and then weighed for flux calculation. After three samples were collected at the 5, 10, and 15 min marks, the feed solution was switched to the desired experimental solution to obtain ultrafiltration data. Three more samples were collected and weighed at the 5, 10, and 15 min marks. After ultrafiltration, the dialysate was turned back on to run dialysis with the desired component. Three flow rates were used to model different rates of dialysis. The first flow rate was 200 mL/min for the feed and 300 mL/min for the dialysate. Feed-in, feed-out, and dialysate-out samples were collected to measure mass transfer. The second flow rate was 300 mL/min for the feed and 500 mL/min for the dialysate. Samples were gathered in the same manner as for the first flow rate. The third flow rate was 400 mL/min for the feed and 500 mL/min for the dialysate. Samples were again collected in the same manner. After samples were collected, the machine was turned off, and the module was disconnected. A 70% ethanol/water mixture was run through the apparatus for 30 min at 500 mL/min to clean and sterilize the system after each use.

### 2.6. Sample Concentration Analysis

The protein samples collected during dialysis and ultrafiltration were analyzed for their concentration using a BioTek Epoch Multi-Volume Spectrophotometer (Winooski, VT, USA). A UV-transparent 96-well plate was utilized. Samples were pipetted in triplicate in 100-µL increments into the well plate. The absorbance was run at 280 nm for the BSA and lysozyme samples. The urea samples were prepared with a QuantiChrom Urea Assay Kit provided from VWR and run at 520 nm in a standard polystyrene 96-well plate, both supplied through VWR.

### 2.7. Scanning Election Microscopy

Scanning Electron Microscope (SEM) images were taken using a Philips XL30 Environmental Scanning Electron Microscope machine (North Billerica, MA, USA) to characterize the membrane morphology. They were taken at 65× and 150× magnification at 10 kV. Cross-sections of the fibers were prepared using a freeze-cracking method.

## 3. Results and Discussion

### 3.1. Modified Membrane Ultrafiltration Performance

A dialysis membrane must be able to operate in ultrafiltration mode (water removal) as well as dialysis mode. This first section will define some of the terms commonly used in dialysis as well as analyzing ultrafiltration performance. One of the first terms used, ***K_UF_***_,_ is the pure water flux of the membrane at a constant pressure, where ***Q_UF_*** is the permeate flow rate, ***A*** is the membrane area, and ***P*** is the pressure on the inlet or outlet of the donating stream [9]. The ultrafiltration mode sieving coefficient (***S_o_***) of each membrane was determined using Equation (2), where ***c_f_*** is the concentration of the filtrate and ***c_p_*** is the concentration of the permeate [9]. It must be noted that two types of sieving coefficients will be used in this paper. The sieving coefficient, as described in Equation (2), is the ultrafiltration sieving coefficient and refers to how much of a component passes through the membrane under a constant pressure, in this case, 1 psi. An element with 0% rejection has a sieving coefficient of 1, and a component with a 100% rejection has a sieving coefficient of 0. Three components of interest—BSA, lysozyme, and urea—were run in ultrafiltration mode to find their sieving coefficients (Figure 3). Appendix A shows the sieving coefficients in table form with standard deviation.
***K_UF_*** = (***Q_UF_***/***A***)/[1/2 (***P_inlet_*** + ***P_outlet_***)],(1)
***S_o_*** = (***c_f_***/***c_p_***),(2)


Sieving coefficient data shows many things about the characteristics of these membranes. First, as would be desired in a dialysis membrane, the control and all modified membranes have urea sieving coefficients near or at unity, ensuring smooth passage of components like urea and salts. Second, the sieving coefficients for the middle molecules range from 0.68 to 0.88. As lysozyme is in the molecular weight range of uremic toxins, it shows that molecules within this size range can pass through the membrane. Finally, the sieving coefficient for BSA under these conditions ranged from 0.13 to 0.31, with the 50/50 membrane having a sieving coefficient significantly higher than any of the other materials. Excessive loss of BSA may make the 50/50 membrane unsuitable for dialysis. To better understand how the additives were affecting the active membrane area, equations were used to estimate their pore sizes. A general set of equations used for calculated pore sizes are given below in Equations (3) and (4) [9]. Equation (3) determines the theoretical solute radius (***R_s_***) of each protein used via the molecular weight (***M_W_***) in Daltons.
***R_s_*** = 3.1 × 10^−11^ (***M_W_***)^0.47752^,(3)
***S_o_*** = (1 − ***λ^2^***) [2 − (1 − ***λ^2^***)] ***exp*** (−0.746 ***λ^2^***)***,***(4)

Equation (4) is a hydrodynamic model that uses the values calculated from the sieving coefficient, where ***λ*** is the ratio of the solute radius (***R_s_***) to the pore radius (***R_p_***) [12]. The pore radius was determined by the minimization of the sum of the squared residuals between the model (Equation (4)) and the data for BSA. The values obtained for pure water flux (***K_UF Pure Water_***), as well as pore size (***R_p_***) along with the area (***A***) in meters squared, hollow fiber radius (***R***) in microns, and hollow fiber thickness (***T***) in microns of the test membranes as measured through SEM are displayed in Table 1. When BSA was used to calculate pore radius, the data suggests that each of the modified membranes had a bigger pore radius than the control with 50/50 showing the most significant increase. The BSA numbers reported were used to estimate pore radius, because BSA is assumed to be spherical while lysozyme is not [26,27,28]. The flux data of these membranes show that the control and Form II are about the same, with about a 10% increase for 50/50 and a 30% increase for Form I. After analyzing all of this data, we conclude that, for pure components in water, it appears from ultrafiltration data that Form I modified is the best candidate, due to its high flux and relatively high rejections of BSA.

### 3.2. Dialysis

Cellulose triacetate membranes are used in dialysis to remove urea, and the rate at which urea is removed is still a major limiting factor in dialysis treatment time. While urea is not considered the most significant component to remove for a patient’s health, it is a good indicator of dialysis membrane performance. (***K_oA_***) is the mass transfer area coefficient, otherwise known as the theoretical clearance at infinite blood and dialysate flow rates. It can be used as a representative term to describe treatment capabilities, as well as to determine theoretical treatment times for patients [29]. (***K_oA_***) is defined using Fick’s law, Equation (5) [30,31]. (***∆C_M_***) represents the change in concentration across the membrane, and (***N***) represents the solute transport rate in mg/min. The solute transport rate (***N***) was calculated using Equation (6), where (***Q_f_***) represents the flow rate, subscript (***f***) represents feed, subscript (***d***) represents dialysate, subscript (***o***) represents out, and subscript (***i***) represents in. (***A***) represents the area, and (***∆C_M_***) represents the log mean concentration difference, which was calculated using Equation (7). Subsequently, the clearance (***K***) can be determined from Equation (8). A series of triplicate dialysis experiments were performed with BSA, lysozyme, and urea, as shown in Table 2.

***N*** = ***K_oA_ A*** (∆***C_M_***),(5)

***N*** = ***Q_f_*** (***C_fi_*** − ***C_fo_***) = ***Q_d_*** (***C_do_*** − ***C_di_***),(6)

**∆*C_M_*** = [(***C_fi_*** − ***C_do_***) − (***C_fo_*** − ***C_di_***)] / ln((***C_fi_*** − ***C_do_***) / (***C_fo_*** − ***C_di_***)),(7)

***K*** = ***N*** / (***C_fo_*** − ***C_di_***)***,***(8)

As can be seen by looking at urea, the clearance (***K_experimental_***) is over 100% higher with Form I as compared to the control. Higher clearance values are also seen for Form II and the 50/50 mix. An increase in clearance was also seen with lysozyme (desirable) and BSA (undesirable) in the modified materials compared to the control. For BSA, the diffusion was calculated based on the sieving coefficient (SC in Table 2), and it ranged from 1×10^−2^ to 8×10^−3^. The value 8×10^−3^ is comparable with the membrane performance of commercial membranes, while the other sieving coefficient values fall slightly outside [11]. However, another study suggests that, for high cut-off membranes, it may fall inside the range, which allows a sieving coefficient to go as high as 0.2 with a protein loss of fewer than 23 grams per session [12].

The process of membrane fabrication depends on the polymeric materials used and their specific interactions with the solvents upon phase separation [32,33]. This is never more evident than when observing the membranes through a Scanning Electron Microscope (SEM). A cross-sectional SEM of each of the four types of membranes is shown in Figure 4. While key parameters of membrane casting, such as spinneret size, bath temperature, uptake speed, drop height, etc., were kept consistent in this experiment, closer inspection with SEM revealed a dramatic change in the internal pore morphologies. Compared to the control (Figure 4a), each of the modified material membranes (Figure 4b–d) showed alterations to the support material that increased porosity caused by the cellulose nanoparticle additive. A similar effect was demonstrated by Bai et al. in the production of PVDF composite membranes using unmodified cellulose nanomaterials, leading to larger finger-like pores and more pores in the bottom surface. As a result, the cellulose nanomaterial composite membranes were improved, i.e., in pure water flux, mechanical properties, and maintained solute rejection [34]. It has been shown that the support structures of membranes can lead to dramatically different separation performances under different operating conditions [35,36]. More importantly, the improvement in the dialysis performance of these membranes can be explained by looking at the backing, i.e., the macroporous support structure. Manickam and McCutcheon derive a theory for transport with pressure-driven versus diffusion-based processes [37]. Their approach shows that backing is a significant source of resistance in a diffusion-based process, while it is not in a pressure-driven process. Thus, more porous, spongy backings would be predicted to lead to a better dialysis transport rate while not necessarily affecting the ultrafiltration rate.

### 3.3. Theoretical Urea Clearance and Treatment Time

While the experiments were conducted with a total volume similar to a dialysis system, the membrane area was approximately an order of magnitude smaller. Therefore, calculations were utilized to determine how these membranes would act if a typical dialysis membrane area were used [38]. The theoretical urea clearance was calculated by simultaneously solving Fick’s law, the log mean concentration difference, and a mass balance to express the ratio ***K***/***Q_f_*** as a function of two dimensionless parameters, neither of which involves solute concentration. This is shown in Equation (9), where ***Z*** is the ratio of the feed flow rate to the dialysate flow rate and ***R*** = ***K_oA_ A***/***Q_f_*** [39,40].
***K***/***Q_f_*** = [1 − exp(***R*** (1 − ***Z***))]/[Z − exp (***R*** (1 − ***Z***))],(9)

The clearance values (***K_Theoretical_***) determined from the above dimensionless expression based on the experimental data can be found in Table 3.

From the theoretical clearance values, predicted treatment time was calculated following the US National Kidney Foundation (***Kt***/***V***) target. In this equation, a 70-kg person, at 60% water weight, must reach a ***Kt***/***V*** of [6]. Governmental standards are set on this premise to ensure patients are successfully treated during their time at the hospital [41]. It was found that Form I had the most drastically decreased treatment time, at a 26% reduction. Form I showed a decrease in treatment time, while the control showed the typical treatment time (approximately four hours on average).

Furthermore, the membrane can also be observed as a function of size, as shown in Figure 5, where the modified membranes show the ability to maintain appropriate treatment times at smaller sizes without a significant loss in performance. In other words, in the control, we see a 25% loss in performance in the control when moving from a 1-m^2^ membrane to one half that size, and, in the modified sample, OTO-CNM Form I mixed-matrix membrane, only an 11% loss in performance. This effect is exacerbated at smaller membrane sizes. Some significant conclusions can be drawn from this data. First, the theoretical treatment time for the 1-m^2^ membrane area is in the range of what has been previously reported [42]. This suggests that these membranes could make effective dialysis membranes. Second, the differences in clearance from the control versus the modified membranes suggest a smaller membrane can be used to achieve the same results seen in the unmodified membranes. It is important to note that results may seem more significant in the smaller test modules than when compared at the one-meter square membrane area. This is because both sets of solutions are using the same volumetric flow rates. Since dialysis is an unsteady process, larger membrane areas have the capacity for high urea removal rates in the beginning, returning the blood at almost 0 urea. This makes the flow rate a major limiting step in the removal of urea. However, when using a smaller module, the membrane can be more fully utilized, and the gradients affecting transfer can remain larger. To fully realize the potential of these new membranes, less membrane area would need to be used in the same operating conditions as traditional hemofiltration membranes. Smaller modules in the same operating conditions allow more effective and efficient treatment with less blood out of the patient. Furthermore, it has been shown that less blood being removed from the body increases the health of the dialysis patient [43,44].

## 4. Conclusions

In this study on the implementation of two derivatives of cellulose nanomaterials into hollow fiber membranes for ultrafiltration and dialysis, OTO-CNM Form I and Form II mixed-matrix membranes have been quantitatively evaluated for the first time by determining model molecule selectivity (BSA, urea, and lysozyme), as well as transport rates (flux and diffusion rates). Each of the three membranes made from cellulose nanomaterials showed improvements when compared to the control (cellulose triacetate). All showed an increase in flux, mass transfer area coefficient, and urea clearance. The differences were more significant when in dialysis mode, which suggests that the porous, spongy backing layer reduced the resistance in the diffusion-based process. These membranes have characteristics that fall within the range of commercial dialysis membranes, indicating that they could be an attractive candidate for future hemofiltration membrane development. Further analysis of dialysis membranes like these will/could show the possibility of the use of less membrane area and shorter treatment times, leading to better patient health.

## 5. Patents

A U.S. patent has been submitted on the OTO-CNM Form I and Form II with a potential royalty stream to the inventors. International Publication Number WO 2019/023702 A1. International Publication Date: 31 January 2019.

## Figures and Tables

**Figure 1 polymers-12-01348-f001:**
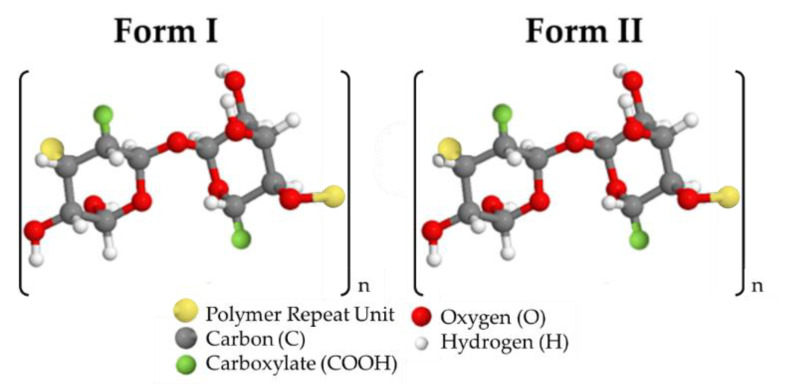
The chemical structure of Form I and Form II OTO-CNM polymers.

**Figure 2 polymers-12-01348-f002:**
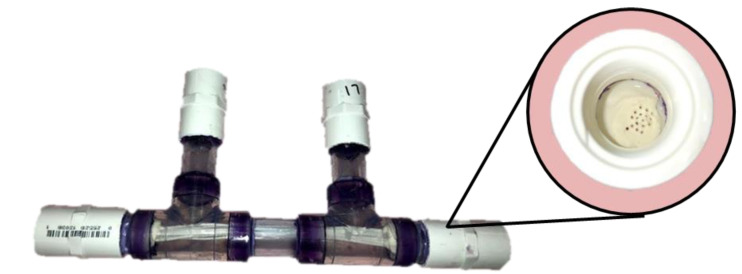
Example of the hemodialysis module used for running tests.

**Figure 3 polymers-12-01348-f003:**
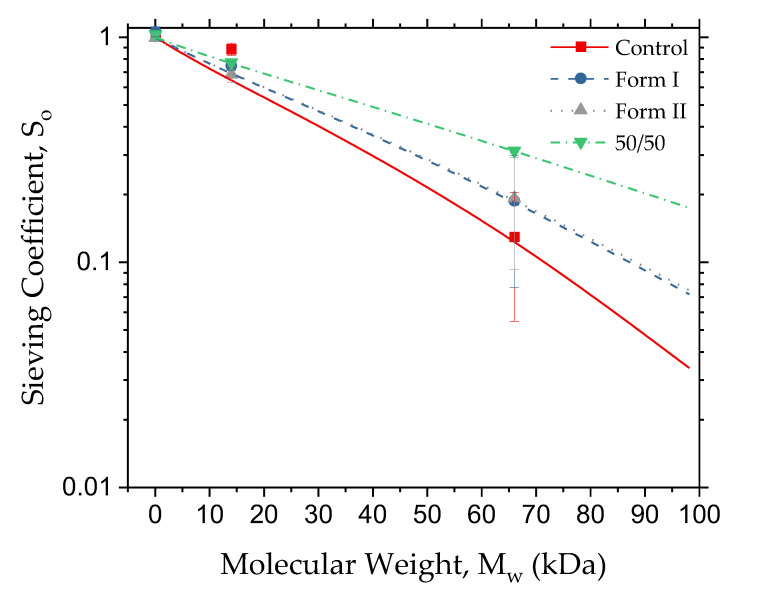
Experimental (points) and theoretical (lines) sieving coefficients (S_o_) plotted on the y-axis, and protein/urea molecular weight plotted on the x-axis.

**Figure 4 polymers-12-01348-f004:**
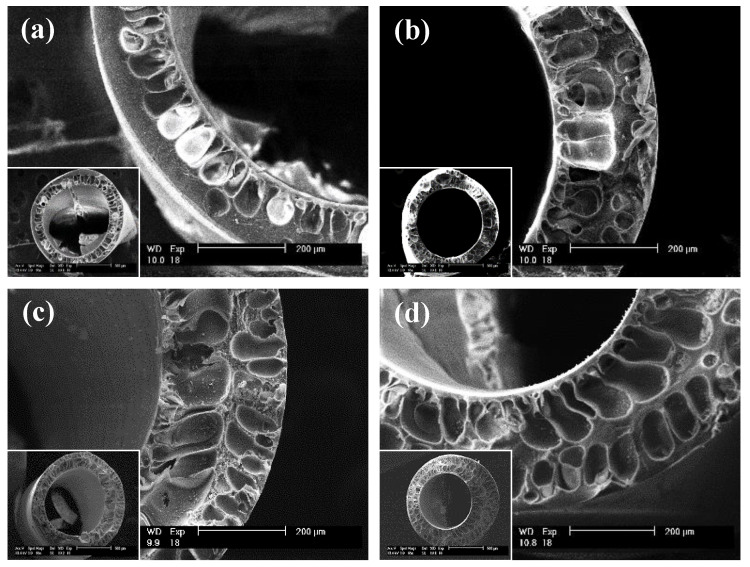
SEM Images of (**a**) the control, (**b**) Form I, (**c**) 50/50, and (**d**) Form II to show pore structure and membrane size at a 200-micron scale.

**Figure 5 polymers-12-01348-f005:**
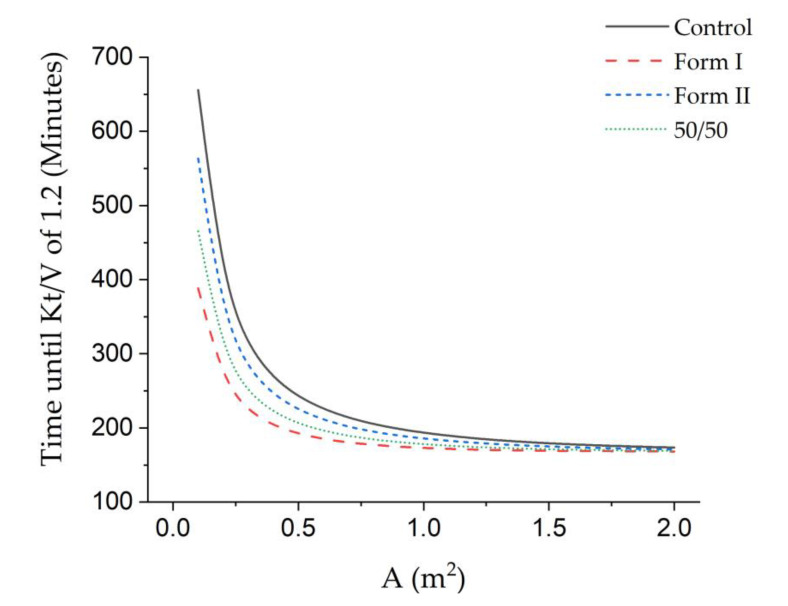
Dimensionless analysis of the time of clinical operation completion (Kt/V = 1.2) as observed against membrane surface area, A (m^2^), for the control, Form I, 50/50, and Form II samples show membrane performance at different active membrane areas under the same conditions, i.e., a 300 mL/min feed flow rate and 500 mL/min dialysate flow rate using the same assumptions utilized in Table 3.

**Table 1 polymers-12-01348-t001:** Experimental values for sample membrane area, radius, thickness, pure water flux, and pore size.

Membrane Material	*A* (m^2^)	*R* (µm)	*T* (µm)	*K_UF Pure Water_* (L/hr/m^2^/PSI)	*R_p_* (nm)
**Control**	0.0115	568	192	109 ± 18	8.9 ± 1.2
**Form I**	0.0112	578	211	144 ± 39	9.8 ± 1.6
**Form II**	0.0118	554	229	111 ± 19	9.9 ± 1.4
**50/50**	0.0105	513	226	121 ± 18	11.6 ± 0.1

**Table 2 polymers-12-01348-t002:** Experimental values for membrane clearance of different molecules, and their respected K_oA_, for the four membrane materials with the bovine albumin sieving coefficient (SC) at a flow rate of 200 mL/min and dialysate flow rate of 300 mL/min.

	Urea		Lysozyme		BSA		
Sample	*K_oA_*	*K_experimental_* (mL/min)	K_oA_	*K_experimental_* (mL/min)	*K_oA_*	*K_experimental_* (mL/min)	*SC*
**Control**	968	9.4	115	1.3	<1	<1	0.0080
**Form I**	1996	20.8	261	2.9	225	2.5	0.0091
**Form II**	1177	14.5	201	2.4	499	5.8	0.0111
**50/50**	1528	16.2	388	4.0	478	5.0	0.0125

**Table 3 polymers-12-01348-t003:** Theoretical urea clearance and treatment times for dialysis using different membrane variations.

Sample	*K_Theoretical_* (mL/min)	*K_Theoretical_* (mL/min/m^2^)	Theoretical Treatment Time (hours)
**Control**	9.2	211	3.98
**Form I**	20.6	284	2.95
**Form II**	12.2	254	3.31
**50/50**	15.9	257	3.26

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
