# Peer review of "Oxone®-Mediated TEMPO-Oxidized Cellulose Nanomaterial Ultrafiltration and Dialysis Mixed-Matrix Hollow Fiber Membranes"

_polymers, 2020, doi:10.3390/polym12061348_

Round 1

Reviewer 1 Report

It is a very good study with overall adequate presentation of experimental results. Some additions are needed:

1) Authors should further emphasize on the novelty of their work.

2) Some minor typos, grammar and syntax errors should be carefully revised and corrected accordingly.

3) Reference can be even more updated (more recent relative works).

4) The major drawback of this paper is the poor discussion. Authors must drastically improve this section (comparison with other papers, explanations, etc)

Author Response

Dear Reviewer 1:

In response to your review, each comment will be addressed directly.

In accordance with comment one (the novelty of the work), we have emphasized that this study is the first to focus on the implementation of OTO-CNMs into hollow fiber membranes and a comparison of these membranes for ultrafiltration and dialysis. Furthermore, we illustrate more clearly throughout the paper that the results presented in this study provide the first quantitative evaluation of the clearance and sieving characteristics of Oxone® Mediated TEMPO-Oxidized Cellulose Nanomaterial doped cellulose triacetate mixed matrix hemodialyzers. I.e., Line 90-92, line 325-328

Next, to address the grammar syntax and minor typos this publication ran through a 3rd party grammar checking software, Grammarly, as well as the paper, having been thoroughly checked, rechecked, and corrected.

Third, several new references in lines, 264, 275, 277, 280 were added to show more relevant work in the area of mixed matrix polymer membranes both with and without cellulose nanoparticulate addition.

Lastly, two of the discussion sections went through major alterations with significant changes made to the SEM section (lines 263-280) of the paper, and the dimensionless analysis (lines 297-310). In the SEM section, the pore morphology was better discussed and supported by up to date references on the effect of additional porosity in the support structure of a membrane. In the Dimensionless analysis, an additional graph was calculated showing membrane performance at different active membrane areas under the same operating conditions, demonstrating the importance of better membranes to achieve smaller effective module sizes.

Reviewer 2 Report

The work reported in this manuscript (Oxone® Mediated TEMPO-oxidized Cellulose Nanomaterial Ultrafiltration and Dialysis Mixed Matrix Hollow Fiber Membranes) is interesting and well presented. However, it needs minor improvements before acceptance. The work requires minor revision. Comments are: ‎

Comment 1: There are some typographical errors in the manuscript, so authors need to correct it in the revised manuscript.

Comment 2: Provide the 'ESEM' full form in the revised manuscript.

Comment 3: In ESEM results: The authors should explore and discuss better their results with some more references in order to prepare a better discussion.

Comment 4: What is the impact of hydrophilic properties? Please explain.

Author Response

Dear Reviewer 2:

In response to your review, each comment will be addressed directly.

In accordance with comment one (typographical errors), this publication was run through a 3rd party grammar checking software, Grammarly, as well as the paper, having been thoroughly checked, rechecked, and corrected.

Second, the full form of ESEM is included in the paper and referenced more appropriately as SEM throughout the paper. This is made clear in the methods section (lines 186-189).

Third, two of the discussion sections went through major alterations with significant changes made to the SEM section of the paper (lines 263-280), and the dimensionless analysis (lines 297-310). In the SEM section, the pore morphology was better discussed and supported by up to date references on the effect of additional porosity in the support structure of a membrane induced through nanoparticulate addition. In the Dimensionless analysis, an additional graph was calculated showing membrane performance at different active membrane areas under the same operating conditions, demonstrating the importance of better membranes to achieve smaller effective module sizes.

Lastly, while hydrophilicity was one of the beneficial properties observed with the nano particulates in our previously published material characterization paper and hydrophilicity can be useful for filtration, the investigation on hydrophilicity was outside the scope of this research, therefore the only mention of hydrophilicity in the paper (in the abstract pg 1 line 14) was removed accordingly.

Round 2

Reviewer 1 Report

All my comments of the initial submission have been correctly replied and included in the revised manuscript. The quality of this work has been drastically improved after revision and therefore I recommend its publication as it is.

Author Response

Dear Reviewer,

In order to check for continuity, grammar, and formatting, the document was passed over to writing specialists at our university for review and correction. Minor edits were made in the form of reference format corrections throughout the publication, general grammar as well as formatting. More specifically an error was found with reference 20-28 in lines 82-93 where references where then all corrected and all future references adjusted and double-checked for correctness. As well as with the image canvas in lines 296-297 where some information had been cut off.  These changes have all been tracked throughout the report.

Sincerely,

John
